# A Novel Multi-Component Formulation Reduces Inflammation In Vitro and Clinically Lessens the Symptoms of Chronic Eczematous Skin

**DOI:** 10.3390/ijms241612979

**Published:** 2023-08-19

**Authors:** Jihee Kim, Eunjoong Jung, Wonmi Yang, Chun-Kang Kim, Serpen Durnaoglu, In-Rok Oh, Chan-Wha Kim, Anthony J. Sinskey, Martin C. Mihm, Ju Hee Lee

**Affiliations:** 1Department of Dermatology & Cutaneous Biology Research Institute, Yonsei University College of Medicine, Seoul 03722, Republic of Korea; mygirljihee@yuhs.ac; 2Scar Laser and Plastic Surgery Center, Yonsei Cancer Hospital, Seoul 03722, Republic of Korea; 3Biocoz Global Korea, R & D Center, Seoul 03181, Republic of Korea; eunjoongjung@biocozglobal.com (E.J.); wonmiyang@biocozglobal.com (W.Y.); chunkangkim@biocozglobal.com (C.-K.K.); serpen@biocozglobal.com (S.D.); inrokoh@biocozglobal.com (I.-R.O.); cwkim@biocozglobal.com (C.-W.K.); 4Department of Biology, Massachusetts Institute of Technology, Cambridge, MA 02139, USA; 5Department of Dermatology, Brigham and Women’s Hospital, Harvard Medical School, Boston, MA 02115, USA; martin.mihm@bwh.harvard.edu

**Keywords:** anti-inflammation, chronic eczema, skin, 3D skin model, cytokine, keratinocytes

## Abstract

Long-term treatments for inflammatory skin diseases like atopic dermatitis or eczema can cause adverse effects. Super Protein Multifunction (SPM) was investigated as a potential treatment for managing skin inflammation by monitoring the expression of pro-inflammatory cytokines induced using LPS and poly(I:C)/TNFα in HaCaT keratinocytes and Hs27 fibroblasts as measured via RT-PCR. SPM solution was also assessed for its effect on cytokine release, measured using ELISA, in a UVB-irradiated 3D human skin model. To evaluate the efficiency of SPM, 20 patients with mild eczematous skin were randomized to receive SPM or vehicle twice a day for three weeks in a double-blind controlled trial. In vitro studies showed SPM inhibited inflammation-induced IL-1β, IL-6, IL-33, IL-1α, TSLP, and TNFα expression or release. In the clinical study, the SPM group showed significant improvements in the IGA, PA, and DLQI scores compared to the vehicle group. Neither group showed significant differences in VAS (pruritus). Histological analysis showed reduced stratum corneum thickness and inflammatory cell infiltration. The results suggest that SPM may reduce inflammation in individuals with chronic eczematous skin.

## 1. Introduction

The skin serves as the body’s outermost layer, protecting it from physical trauma, toxins, and microbes. The integrity of the skin tissue is maintained by proliferating basal keratinocytes, which give rise to an epithelial layer of differentiated keratinocytes, which then transforms into the outermost cornified layer composed of dehydrated cells and keratin filaments, forming a mechanical barrier against the entry of foreign agents, and protecting the body from loss of hydration [1,2]. This mechanical protection is complemented by an epithelial immune system in which keratinocytes secrete antimicrobial peptides that control microbial growth and cytokines that recruit lymphocytes to the site of injury or infection to destroy and remove foreign pathogens [3,4,5,6]. Normal inflammatory responses maintain skin homeostasis. However, in cutaneous inflammatory diseases like atopic dermatitis, uncontrolled pro-inflammatory signals from the skin tissue may integrate with systemic inflammatory processes to cause bouts of excessive cutaneous immune responses that alter the skin architecture and damage the integrity of this mechanical barrier. These effects manifest as a keratinocyte-derived cytokine signature consisting of erythema and edema with accompanying xerosis and loss of the skin barrier function [7,8,9,10,11]. 

SPM, or Super Protein Multifunction, is a composite formulation that consists of essential components required by cells to grow and survive in vitro, enhancing their recovery capability [12]. The proprietary SPM formulation includes the following components, in order of relative abundance: sodium chloride, sodium bicarbonate, recombinant human serum albumin, L-glutamine, potassium chloride, L-arginine hydrochloride, calcium chloride, L-lysine hydrochloride, sodium phosphate monobasic, sodium pyruvate, L-threonine, L-valine, L-histidine hydrochloride, L-serine, glycine, L-cysteine hydrochloride, L-proline, L-methionine, myo-inositol, L-alanine, thiamine hydrochloride, niacinamide, and pyridoxine hydrochloride. The components within SPM have the potential to improve skin health by reducing inflammation. Vitamins such as thiamine (vitamin B1), niacinamide (vitamin B3), pyridoxine (vitamin B6), and i-inositol are precursors of metabolic enzyme cofactors or co-enzymes, which are crucial for cell metabolism and associated with promoting skin health [13,14,15,16,17,18]. HSA in the human bloodstream functions as a carrier protein that influences the stability and bioavailability of hydrophobic molecules. As a major contributor to plasma oncotic pressure, HSA is included in some cell culture media [19] and drug formulations [20,21]. HSA solutions are also used as intravenous volume-expanding resuscitation fluids [22]. Interestingly, the clinical benefits of HSA injection have been observed to be greater than those of other volume expanders for treating cirrhosis, peritonitis, and sepsis [23,24,25,26,27], suggesting that HSA may play an anti-inflammatory role that is independent of its volume-expanding function [28]. 

Additionally, certain amino acids have been shown to curtail cutaneous inflammation. Glycine can activate glycine-gated chloride channels to hyperpolarize keratinocytes and reduce inflammatory output [29,30]. Arginine supplementation has been reported to attenuate lipopolysaccharide (LPS)-induced NF-κb activation and cytokine expression in mammary epithelial cells, intestinal epithelial cells, and dermal fibroblasts challenged with LPS, a unique component of Gram-negative bacteria and a Toll-like receptor 4 (TLR4) agonist [31,32,33,34]. Current treatments for atopic dermatitis and eczema include emollients, which physically supplement a variety of topically or orally administered steroids and calcineurin inhibitors, or subcutaneously injected biologics that pharmacologically target local and systemic inflammation mediators [35]. However, prolonged use of topical steroids may cause adverse effects, such as erythema, hypertrichosis, perioral dermatitis, photosensitivity, and burning sensation [36]. Therefore, there is a need for alternatives such as non-steroid to steroid creams. In this study, we aimed to evaluate the potential of SPM as a conventional non-steroidal anti-inflammatory treatment for managing cutaneous diseases. We first investigated the effects of SPM supplementation on the cellular innate immune response triggered by LPS and poly(I:C)/TNFα in HaCaT keratinocytes and Hs27 fibroblast cell lines by measuring the levels of pro-inflammatory cytokines. We then explored whether the results are recapitulated in a UVB-irradiated 3D human skin model. Finally, to evaluate the potential translation of in vitro findings to the clinic, we conducted a study to investigate whether the topical application of SPM solution could effectively modulate the signs and symptoms of eczematous skin lesions in human patients. 

## 2. Results

### 2.1. SPM Treatment Suppresses Inflammatory Response Induced by LPS

LPS induces inflammation in human skin cells and stimulates a robust immune response by inducing inflammatory cytokines, including tumor necrosis factor α (TNF-α), interleukin-1 family (IL-1), and interleukin-6 (IL-6) [37]. LPS also reduces the viability of epithelial cells [38,39]. The effect of LPS on human keratinocytes depends on its primary receptor, TLR4 [40,41]. TLR4 is also a component of the inflammasome complexes involved in the activation of interleukin-1β (IL-1β) in humans [42]. We treated human HaCaT keratinocytes with LPS at concentrations of 0 (control), 10, 25, or 50 μg/mL for 24 h. Cell viability was significantly reduced in cells exposed to 25 or 50 μg/mL LPS (16.9 and 27.14%, respectively) compared to the control group (Figure 1a). To investigate the potential protective effects of SPM against LPS-induced inflammation, we exposed cells to LPS at concentrations of 25 or 50 μg/mL for 24 h, followed by treatment with 10% or 20% SPM for an additional 24 h. Different SPM doses (ranging from 10% to 50%) were previously evaluated in vitro, and 20% SPM showed the highest viability. Therefore, it was selected as the optimal dose for cell culture. SPM notably reversed the reduction of cell viability induced by LPS, and cell viability increased by 11.2% and 23.8%, respectively (Figure 1b). These results suggested that SPM preserves cell viability when there is LPS-induced damage and may help cell regeneration.

To investigate the effect of SPM on inflammation, we assessed the IL-1β mRNA expression in HaCaT cells after a 24 h stimulation with LPS, followed by treatment with 10% or 20% SPM. The objective was to compare the optimal dose (20%) with the lowest dose (10%) of SPM to determine whether even the lowest dose exhibits an anti-inflammatory effect. First, cells were stimulated with 0 (control), 25, or 50 μg/mL LPS for 24 h. The expression of IL-1β was significantly increased by LPS at the concentration of 50 μg/mL (Figure 2a). In unstimulated cells, IL-1β expression was not affected by SPM treatment (Figure 2b). However, IL-1β levels were significantly suppressed by SPM (20%) compared to control cells after stimulation with 50 μg/mL LPS (Figure 2c).

In human epithelium-derived fibroblasts, LPS induces the production and secretion of cytokines IL-1α and IL-6 [43,44,45,46,47,48]. Therefore, we checked the effect of LPS on cell viability in Hs27 human fibroblast cells. Cells were exposed to LPS at concentrations of 0 (control), 50, 100, or 150 μg/mL for 24 h. Cell viability was significantly reduced by each dose of LPS (Appendix A). To examine the LPS-induced cytokine levels in Hs27 cells, we exposed cells to 100 μg/mL LPS. LPS exposure induced the expression of IL-1α and IL-6, which remained elevated compared to unstimulated controls (no LPS) even 24 h after LPS removal (Appendix A), indicating a sustained inflamed state. Treatment with 20% SPM reduced IL-1α and IL-6 induction compared to cultures supplemented with the vehicle (Appendix A), suggesting that SPM supplementation limited cytokine secretion in inflamed Hs27 cells.

These results indicate that SPM supplementation suppresses cytokine expression from inflamed skin-derived epithelial and fibroblast cultures.

### 2.2. SPM Treatment Downregulates the Expression of Cytokines Induced by Poly (I:C)/TNFα in HaCaT Cells

We evaluated SPM’s anti-inflammatory effects by measuring the cytokine levels (IL-1β, IL-6, IL-33, and TSLP; Thymic stromal lymphopoietin) in HaCaT cells stimulated by poly(I:C)/TNFα (Figure 3a,b). Poly(I:C) (polyinosinic-polycytidylic acid) is a double-stranded RNA analog that activates TLR3, eliciting cytokine release in primary human keratinocytes and HaCaT cells [49,50]. TNFα has a master regulatory role in cytokine production and potentiates cytokine outputs triggered by other pro-inflammatory stimuli in HaCaT cells [51,52,53]. Treatment with an inflammatory cocktail (10 μg/mL poly(I:C) and 20 ng/mL TNFα) along with a vehicle (water) for 24 h upregulated the expression of IL-1β, IL-6, IL-33, and TSLP compared to the expression levels in the control cultures. Co-treatment with poly(I:C)/TNFα and 20% SPM for 24 h showed lower cytokine expression than only poly(I:C)/TNFα treated cultures (Figure 3b), suggesting that SPM supplementation suppressed the expression of IL-1β, IL-6, IL-33, and TSLP. Compared with the positive control (1 μM mometasone), SPM had similar effects. These results demonstrate that SPM supplementation limits cytokine output from inflamed keratinocytes in vitro.

### 2.3. SPM Suppresses the UVB-Induced Production of Inflammatory Cytokines in a Human Epidermis Model

We also investigated the anti-inflammatory effect of SPM on UVB-induced inflammation in a 3D human skin model; higher doses of SPM (50% and 100%) were tested to ensure a measurable response within the model. After exposure to 250 mJ/cm^2^ UVB, 3D epidermis (KeraSkin^TM^) was treated for 8 h with SPM or topical corticosteroids (HC; hydrocortisone, MF; mometasone furoate) as positive controls. Then, treatment agents were rinsed off with PBS, and the skin culture was maintained for another 24 h (Figure 4a). The basal culture media was collected to detect secreted cytokines using ELISA at 8 h after treatment and 24 h after recovery. UVB irradiation highly increased the secretion of IL-1α, TSLP, and TNFα compared to levels observed in unirradiated skin (no UVB) (Figure 4b–d). Treatment with 50% or 100% SPM for 8 h after UVB exposure limited the release of IL-1α, TSLP, and TNFα compared to untreated cultures irradiated with UVB alone. After 24 h of recovery, IL-1α and TNFα levels remained lower than those observed for UVB-irradiated controls. These results were consistent with the anti-inflammatory effect of SPM seen in vitro. These data, taken together, demonstrate that SPM inhibits UVB-induced production of inflammatory cytokines in the epidermis.

### 2.4. SPM Ameliorates Symptoms and Signs of Eczematous Dermatitis

Unmitigated cutaneous immune responses are integral to local and systemic inflammatory processes that manifest in dermatological ailments [9,10,11]. Cutaneous inflammation and partial regeneration of the skin barrier underlie the pathogenesis of eczema and atopic dermatitis and are points of control in their treatment [35,54]. To examine the anti-inflammatory effects of topically applied SPM on skin lesions associated with eczema and dermatitis, we conducted a randomized, double-blind, placebo-controlled clinical study.

Twenty subjects with eczematous skin diseases such as chronic eczema or atopic dermatitis (Table 1) were entered into the study. Adjacent or equivalent (i.e., left arm and right arm) skin lesions were topically treated twice a day with humectant containing SPM (SPM) or the humectant alone (vehicle) as an in-patient placebo control (see Appendix A). After three weeks of treatment, the severity of eczema symptoms—erythema, scaling, excoriation, edema, lichenification, and papules—was assessed, either firsthand (investigator global assessment, IGA) or from photographs of the skin lesions (photographic assessment, PA). At 3 weeks of treatment, SPM significantly increased the IGA score (vehicle group; 1.85 ± 0.81, SPM group; 2.80 ± 0.95), indicating an improvement in clinical symptoms (*p* < 0.05) (Figure 5a), and reduced the PA score (vehicle group 2.8 ± 0.77 (D0) to 2.4 ± 0.80 (D21), SPM group 2.85 ± 0.79 (D0) to 1.25 ± 0.44 (D21)) of disease severity compared to vehicle (*p* < 0.05) (Figure 5b). The Dermatology Life Quality Index (DLQI) score showed a significant reduction (from 6.05 ± 3.67 to 1.45 ± 1.03) after 3 weeks of SPM treatment (*p* < 0.05) (Figure 5c). The subjects also reported reduced itchiness at both SPM-treated and vehicle-treated skin lesions (Figure 5d), suggesting that controlling eczema and dermatitis symptoms in one skin lesion may be sufficient to tone down pruritus beyond the immediate area of the treated skin lesion. Histologic analysis of one patient’s skin lesion (Figure 6a) showed increased stratum corneum thickness before SPM application and perivascular lymphocyte infiltration, which were reduced after 3-weeks of application of SPM (Figure 6b,c). No adverse reactions were reported during the three-week study. Taken together, the improvements in clinically assessed skin lesions treated with SPM and the reported enhancements in the quality of life by study subjects support the effectiveness of SPM in ameliorating the clinical symptoms of eczematous skin lesions.

## 3. Discussion

In this combined in vitro and clinical study, we have shown the regenerative and anti-inflammatory effects of SPM. Cutaneous inflammation contributes to progressive symptoms of eczema and chronic atopic dermatitis (AD) [35]. IL-1α, IL-1β, and IL-6 are acute immune response cytokines that drive inflammation [55,56] by inducing the expression of chemokines; the cytokines, in turn, attract effector T-cells [57]. TNFα is a major inflammatory effector [58]. TSLP and IL-33 are elevated in AD patients [59], and IL-33 stimulates the typical Th2 cytokine profile, a characteristic of AD inflammation [60,61]. These cytokines are upregulated in inflamed lesions in rheumatoid arthritis, burns, AD, or psoriasis and have become targets of drugs developed to treat these ailments [55,56,62,63,64]. 

In our in vitro studies, SPM supplementation suppressed IL-1β, IL-6, IL-33, and TSLP transcription in inflamed human keratinocytes (Figure 2 and Figure 3) and IL-1α and IL-6 transcription in human fibroblasts (Appendix A). In addition, IL-1α, TSLP, and TNFα secretion was limited by SPM in a UVB-irradiated 3D human epidermis model (Figure 4b–d). Our results suggest that SPM can limit skin inflammation at the cellular (Figure 2 and Figure 3) and tissue (Figure 4) levels.

In our clinical study, 3 weeks of topical SPM treatment on eczematous skin lesions ameliorated the signs and symptoms of eczema and atopic dermatitis (Figure 5a–d). Although the histological analysis of the biopsy sample was limited to one, a reduction in lymphocyte infiltration was observed at the interface between the epidermis and dermis (Figure 6a–c).

SPM comprises numerous components with potential or reported bioactivity in the skin tissue. Among the active components of SPM is an array of amino acids; these can activate specific amino acid receptors to influence signal transduction, proliferation, and inflammation [30,32,65,66]. On the other hand, some amino acids can also enter keratinocytes through specific transporters to affect cellular behaviors [67]. The regenerative effects of vitamins for skin tissue maintenance are well documented [13,18,68,69], but their mechanisms of action are unclear. Currently, the best explanation is that vitamins play a non-specific permissive role by enhancing cellular repair.

In addition to immunological dysfunction, disturbed skin barrier also plays a role in developing AD and eczema [70]. Patients with AD have reduced levels of epidermal differentiation proteins associated with skin barrier dysfunction, such as filaggrin, loricrin, and involucrin [71]. The downregulation of E-cadherin, which is required for tight junction formation, is correlated with filaggrin insufficiency in AD [61] and is shown in eczematous dermatitis [72]. Thus, our results could be complemented by using expression analysis of skin barrier-related proteins.

In our UV-irradiated 3D epidermis model, an increased level of p63 in the presence of SPM was observed p63, a keratinocyte stem cell marker [73,74] helps regulate the growth and differentiation of epidermal stem cells (EPSCs) and is involved in wound healing [75,76]. A recent study also reported that ΔNp63 is critical in the adult epidermis to suppress inflammatory cytokines associated with inflammatory diseases such as AD and psoriasis [77].

One could speculate that SPM inhibits inflammation and increases regeneration by promoting stem cell proliferative potential and activation. However, how SPM relates to stem cell activation requires further exploration. 

Future studies of topical SPM treatment on eczema or atopic dermatitis with various degrees of disease, longer treatment durations, and in combination with other treatments such as steroids will help us determine the best practice for incorporating SPM to treat inflammatory skin diseases.

In conclusion, SPM supplementation significantly reduces the expression and release of inflammatory cytokines in human skin cells in vitro and in a 3D epidermis model. When applied topically, SPM limits the symptoms of eczema and atopic dermatitis for at least three weeks. Our findings suggest that SPM may be developed as a potentially effective topical agent for managing inflammation in case of mild or moderate cutaneous lesions such as eczematous dermatitis.

## 4. Materials and Methods

A detailed description of the materials and methods is provided in Appendix A. The primers used are detailed in Appendix A.

### 4.1. Cell Culture

HaCaT cells were obtained from CLS Cell Lines Service GmbH (Cat # 300493), and Hs27 cells were purchased from ATCC, American Type Culture Collection (CRL-1634). Dulbecco’s Modified Eagle’s Medium (DMEM) was purchased from Gibco (Cat # 11996-065) and ATCC (Cat # 30-2002). HaCaT cells were grown in DMEM (Gibco, Waltham, MA, USA) supplemented with 10% fetal bovine serum (Cat # 16000-044, Gibco), 100 U/mL penicillin, and 100 mg/mL streptomycin (Cat # P4458, Gibco) in 5% CO_2_ at 37.5 °C. Hs27 cells were grown in DMEM (ATCC) with 10% fetal bovine in the same conditions. The cells were removed from the wells with 0.25% trypsin EDTA for each passage. 

### 4.2. Cell Viability Assay

The viability of HaCaT cells was determined by MTT (3-(4,5-dimethylthiazol-2-yl)-2,5-diphenyltetrazolium bromide) assay (Thermo Fisher, Waltham, MA, USA). The viability of Hs27 fibroblasts was determined via automated time-course imaging using Incucyte (Sartorius, Göttingen, Germany).

### 4.3. Cell Stimulation

To induce inflammatory stimulation, we seeded HaCaT and Hs27 cells in 6-well plates and cultured them for 24 h to reach 50% confluency. The culture medium was removed and replaced with serum-free DMEM supplemented with LPS or 10 μg/mL poly(I:C) and 20 ng/mL TNFα. Cells were incubated with LPS for 24 h. After washing with PBS twice, they were treated with SPM or vehicle (distilled water in serum-free DMEM) for another 24 h. Cells were incubated together with poly(I:C)/TNFα cocktail and SPM for 24 h.

### 4.4. Human Skin Model

The growth and differentiation of KeraSkin^TM^, a human 3D epidermis, from primary keratinocytes were carried out by Biosolution (Seoul, Republic of Korea). Fully grown KeraSkin^TM^ samples were irradiated with 250 mJ/cm2 UVB, and the apical surface of the skin model was treated with the following agents for 8 h: 50% SPM, 100% SPM, 60 μM hydrocortisone lotion (HC), or 10 μM mometasone furoate (MF). HC and MF were used as positive controls. The basal culture media was then collected to detect secreted proteins using ELISA.

### 4.5. Enzyme-Linked Immunosorbent Assay (ELISA)

ELISA kits (R & D systems, Santa Clara, CA, USA) were utilized to measure IL-1α, TSLP, and TNFα in a culture medium. The content was measured following the provided protocol.

### 4.6. Statistical Analysis

GraphPad 10.0.1(218) software was used to analyze statistical analyses of in vitro studies. All data points were presented as means and standard deviations. One-way ANOVA followed by Tukey’s multiple comparison tests was used to determine statistical differences between groups. The statistical significance was determined by *p*-values < 0.05.

### 4.7. Clinical Study

The study was a three-week, randomized, double-blinded, placebo-controlled clinical trial conducted at Yonsei University Severance Hospital (Seoul, Republic of Korea) from October 2018 to February 2019. The protocol was approved by the Yonsei University Severance Hospital Institutional Review Board (IRB protocol no. 4-2018-0124). A total of 23 subjects were screened, and 20 subjects (16 females, 4 males) with Fitzpatrick skin types II-IV enrolled in the study after their written informed consent was approved by the IRB. Inclusion criteria included the following (Table 1). Eligible participants were men and women aged 19 years or older (23 to 80 years of age) with chronic eczema, xerosis cutis, atopic dermatitis, or seborrheic dermatitis; subjects who were available to make the return visits (V1 and V2) during the clinical trial; and subjects who could carry out the required tasks for the clinical trial (expressed in a written consent). Exclusion criteria were infections at the skin lesion, acute or chronic illnesses that may have affected the test results, the use of steroid-containing skin cosmetics for the treatment of skin diseases for more than 2 weeks, unable to comprehend the objectives of the clinical tests or carry out the components of the instructions necessary for treatment and study, pregnancy, breastfeeding, or planning a pregnancy during the study period. 

Subjects were randomly assigned to either a vehicle or an SPM solution. Study subjects were instructed to topically apply either the SPM test solution or the placebo (with vehicle only) exclusively to each randomly designated skin lesion with 1 mL dosage volume per single palm area of skin lesion twice daily (once in the morning and once in the evening). Subjects were also instructed to refrain from local treatments, such as skin peeling and laser treatment, and from using cosmetics containing growth factors or stem cell culture fluids on the application area during this study. The test and vehicle agents were randomly coded, and the subjects and the clinical research staff carrying out the administration, training, and distribution were blind to the contents of the test agents. 

During self-administration, the subjects returned to the clinic twice: 1 week (±3 days) after treatment initiation for mid-treatment evaluation and 3 weeks (±3 days) after treatment initiation for end-of-treatment evaluation. A dermatology specialist evaluated the skin lesions to which SPM or vehicle had been applied by Investigator Global Assessment (IGA) by scoring the changes in signs or symptoms (erythema, scaling, excoriation, edema, lichenification, and papules) as follows: 1, no improvement; 2, 1–25%; 3, 26–50%; 4, 51–75%; or 5, 76–100% at the end of the study, and via photographic assessment (PA). For PA, two investigators independently evaluated the severity of skin lesion symptoms on a scale of 1 to 5 as they appeared in digital photographs of the skin lesions taken at the beginning and the end of the 3-week treatment. Subject-reported impact on quality of life from the skin symptoms was evaluated from the Dermatological Life Quality Index (DLQI), in which a lower score correlates with a more minor impact at the end of the 3-week application. The degree of itchiness for the SPM or vehicle-applied skin lesion was evaluated using VAS (0–10 scale, lower score correlates to less itch) within the last 24 h from the time of reporting. DLQI and VAS questionnaires were conducted at the beginning and the end of the 3-week treatment. The investigators, board-certified dermatologists, were kept blind to the treatment groups until the end of the study.

Outside of subject visits, any adverse reactions during the treatment or within 30 days from the end of treatment were to be reported by the subjects within 24 h of occurrence, and causality between the adverse reactions and the treatment was assessed by the investigator.

### 4.8. Statistical Analysis of Clinical Data

IBM SPSS Statistics for Windows version 23.0 was used to perform statistical analyses (IBM Corp., Armonk, NY, USA). Mann–Whitney U tests were used to test for statistical significance. The study results were accepted as statistically significant if *p* < 0.05.

## Figures and Tables

**Figure 1 ijms-24-12979-f001:**
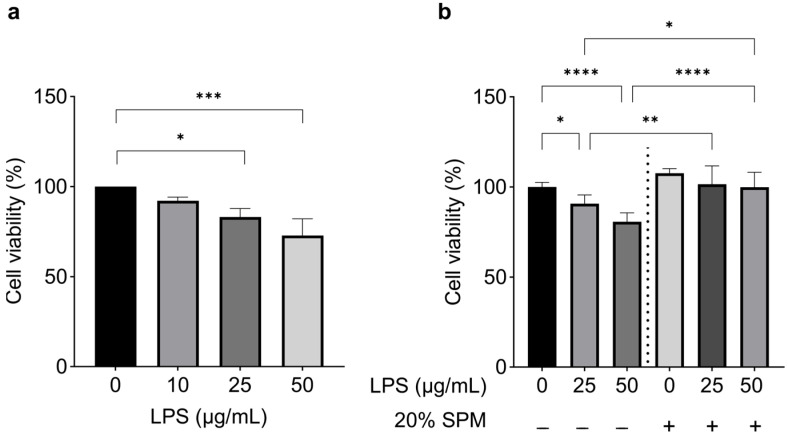
SPM reverses LPS-induced damage. (**a**) Viability of HaCaT cells assessed using MTT assay after 24 h exposure to varying concentrations of LPS (0, 10, 25, or 50 μg/mL). (**b**) Viability of HaCaT after exposure to LPS (0, 25, or 50 μg/mL) followed by treatment with 20% SPM. Error bars represent the mean ± SD. *N* = 3 biological replicates. * *p* < 0.05; ** *p* < 0.01; *** *p* < 0.001; **** *p* < 0.0001, as determined via One-way ANOVA followed by Tukey’s multiple comparison test.

**Figure 2 ijms-24-12979-f002:**
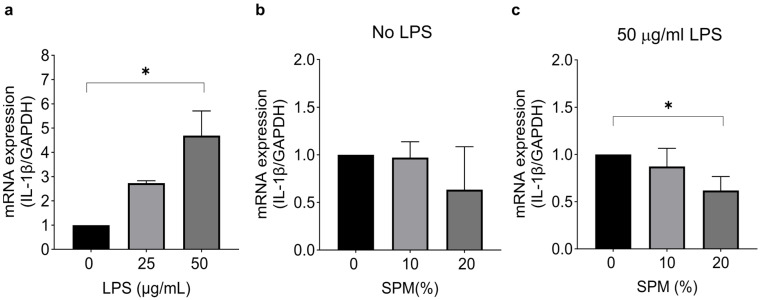
SPM suppresses the expression of LPS-induced IL-1β in HaCaT cells. (**a**) HaCaT cells were stimulated with 25 or 50 μg/mL LPS for 24 h. RT-qPCR was used to determine mRNA expression levels. The mRNA expression level of IL-1β was normalized to GAPDH mRNA expression level. (**b**) Without LPS stimulation, cells were treated with 0% (vehicle control; distilled water in media), 10%, and 20% SPM for 24 h. (**c**) After 24 h of LPS treatment, LPS was removed, and cells were treated with 0% (vehicle control; distilled water in media), 10%, and 20% SPM for another 24 h. Error bars represent the mean ± SD. *N* = 3 biological replicates. * *p* < 0.05, as determined via One-way ANOVA followed by Tukey’s multiple comparison test.

**Figure 3 ijms-24-12979-f003:**
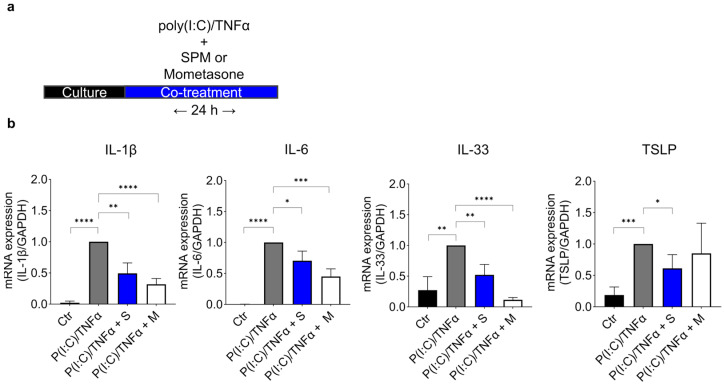
SPM suppresses the expression of poly(I:C)/TNFα-induced cytokines in HaCaT cells. (**a**) Schematic experimental design of 24 h poly(I:C)/TNFα pro-inflammatory cocktail stimulus accompanied by SPM or Mometasone (positive control). (**b**) IL-1β, IL-6, IL-33, and TSLP mRNA levels after 24 h poly(I:C)/TNFα stimulation and co-treatment with SPM or Mometasone. *N* = 6 biological replicates. Error bars represent the mean ± SD. * *p* < 0.05; ** *p* < 0.01; *** *p* < 0.001; **** *p* < 0.0001, as determined via One-way ANOVA followed by Tukey’s multiple comparison test. Ctr; control, P(I:C)/TNFα; poly(I:C)/TNFα + vehicle(water), P(I:C)/TNFa + S; poly(I:C)/TNFα + 20% SPM, P(I:C)/TNFα + M; poly(I:C)/TNFα + 1 μM Mometasone.

**Figure 4 ijms-24-12979-f004:**
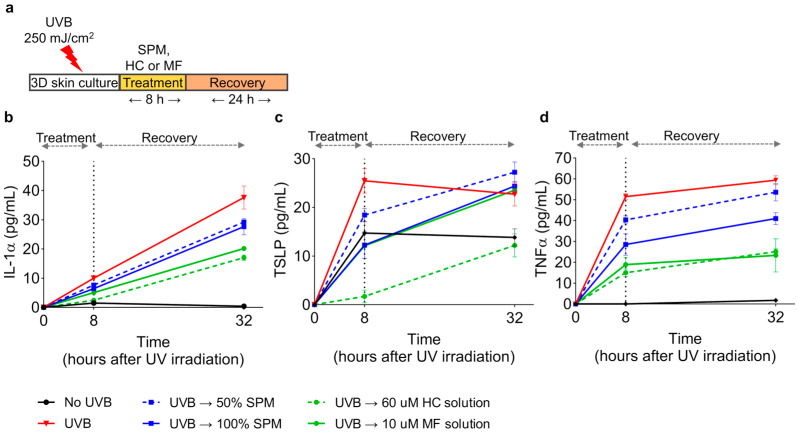
SPM suppresses cytokine secretion in UV-damaged cells in a 3D human epidermis model. (**a**) Schematic of UVB (250 mJ/cm^2^) irradiation and 8 h of treatment with indicated agents following 24 h of culture maintenance. Supernatants were harvested at 8 h and 32 h following UVB irradiation, and the cytokine concentrations were determined using ELISA. (**b**–**d**) Secretion levels of cytokines, IL-1α, TSLP, and TNFα after UVB irradiation. HC (hydrocortisone) and MF (mometasone furoate) were used as positive controls. *N* = 3 biological replicates. Error bars represent the mean ± SD.

**Figure 5 ijms-24-12979-f005:**
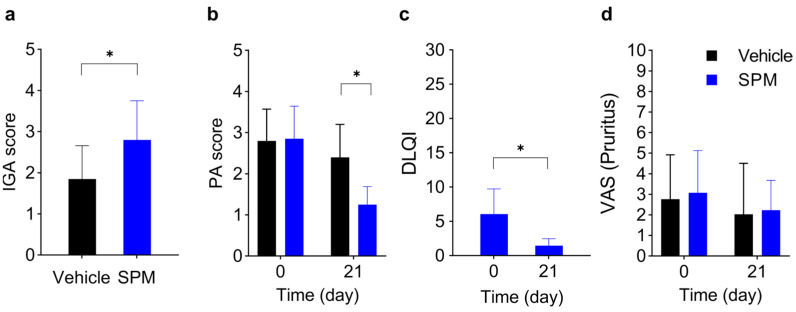
The topical agent containing SPM alleviates mild chronic atopic dermatitis and eczema symptoms. (**a**) Investigator global assessment (IGA), (**b**) photographic assessment (PA) scores (from 1 to 5), (**c**) dermatological quality of life index (DLQI) scores (from 0 to 30), (**d**) visual analog scale (VAS) scores (from 1 to 10) for pruritus (itch) of skin lesions treated with SPM containing topical agent (SPM) and the same topical agent without SPM (vehicle) between day 0 and day 21. Subjects reporting no pruritus at the beginning of the study were excluded from the analysis (*N* = 14). (**a**–**d**) ANCOVA with Mann–Whitney U-test, * *p* < 0.05, *n* = 20 unless noted otherwise.

**Figure 6 ijms-24-12979-f006:**
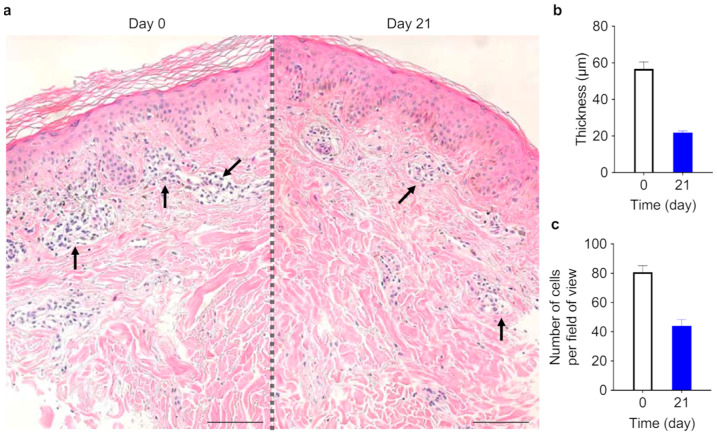
Corneum stratum and lymphocyte infiltration at the dermis–epidermis interface in skin lesions of one subject before and after topical application of SPM. (**a**) Histopathological examination of treated skin as assessed by hematoxylin-eosin (H & E) staining. Lymphocyte infiltration (pockets of nuclei, black arrows) at the dermis-epidermis interface. 200× objective. Scale bar is 100 μm. (**b**) Stratum corneum thickness before and after 3 weeks of SPM treatment. (**c**) Counts of lymphocytes infiltrating the epidermis–dermis interface before and after 3 weeks of SPM treatment.

**Table 1 ijms-24-12979-t001:** Demographic data for the subjects.

Total Number of Patients	20
Female	16
Male	4
Mean age Ethnicity Fitzpatrick Skin Type	43 ± 14 (23~80) Korean II~III
Disease distribution; Xerotic eczema/xerosis cutis Atopic dermatitis Seborrheic dermatitis	14 4 2

## Data Availability

Not applicable.

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
