# Peer review of "A Novel Multi-Component Formulation Reduces Inflammation In Vitro and Clinically Lessens the Symptoms of Chronic Eczematous Skin"

_ijms, 2023, doi:10.3390/ijms241612979_

Round 1

Reviewer 1 Report

The article is a good one, to be picky it only needs a minor revision  and the addition of a few positive controls in some experiments, to improve the research design. Introduction provide sufficient background and include all relevant reference. The  results are clearly presented and the conclusions are supported by results. Good English  

1) line 78: Perhaps do you mean the contrary?

2) line 104 and Fig 2 you have the possibility to introduce also a positive controll (mometasone)

3) missing title "FIGURE 3"

4) line 142  and Fig 3 : stimulation only with poly (I:C)/TNF or poly I:C/ TNF + vehicle water? 

5) line 126 and figure S1: missing positive controll

6) line 377 : "supporting information ca be downloaded at" ?

7) please explain  how the percentage of SPM used in the various experiments was chosen

Author Response

Thank you very much for your precious comments that helped us improve this manuscript. Below we provide the point-by-point responses. 

1) line 78: Perhaps do you mean the contrary?

A: Thank you very much for pointing this out.

We revised the sentence as follows in line 78-79:

“In this study, we aimed to evaluate the potential of SPM as a conventional non-steroidal anti-inflammatory treatment for managing cutaneous diseases.”

2) line 104 and Fig 2 you have the possibility to introduce also a positive control (mometasone)

A: Initially, positive controls were not included in the LPS stimulation experiments because the focus was on investigating the anti-inflammatory effect specifically in response to LPS. As we shifted to poly(I:C)/TNFa as the main inflammation stimulator, which has distinct characteristics and responses compared to LPS, positive controls were included to compare our findings with commonly used topical corticosteroids.

3) missing title "FIGURE 3"

A: We apologize for the omission during the manuscript formatting. The title for figure 3 has been added on line 161.

SPM suppresses IL-1β, IL-6, IL-33, and TSLP cytokine expression induced by poly(I:C)/TNFα cocktail in HaCaT cells” was the original title, but due to potential confusion, it was changed to “SPM suppresses the expression of poly(I:C)/TNFα-induced cytokines in HaCaT cells.”

4) line 142 and Fig 3 : stimulation only with poly (I:C)/TNF or poly I:C/ TNF + vehicle water?

A: To address to the reviewer’s comment to clarity the experimental condition, the stimulation was done using poly(I:C)/TNFa along with a vehicle (water). This information was added to the manuscript line 152-154:

“Treatment with an inflammatory cocktail (10 ug/mL poly(I:C) and 20 ng/mL TNFα) along with a vehicle (water) for 24 h upregulated the expression of IL-1β, IL-6, IL-33, and TSLP compared to the expression levels in the control cultures.”

5) line 126 and figure S1: missing positive control

A: Thank you for your observation regarding the absence of positive controls in our experiments. We acknowledge this point and want to clarify that our study was specifically designed to assess the direct impact of SPM against LPS-induced inflammation. The exclusion of positive controls aligns with this focus, allowing us to concentrate on the novel multi-component formulation's effects.

6) line 377 : "supporting information ca be downloaded at" ?

A: Thank you for noticing the missing information in line 377 regarding the download location for supporting information. At this current stage, all the supporting materials are included in the supplementary files for the reviewers' convenience.

Upon final acceptance, we will add the specific URL where the supporting information can be downloaded by readers. The line 385-386 will then read:

"Supporting information can be downloaded at [insert appropriate URL after acceptance: at www.mdpi.com/XXXX-XXXX]."

7) please explain how the percentage of SPM used in the various experiments was chosen

A: We appreciate your interest in our methodology.

The rationale and details for choosing the specific percentages of SPM was considered based on MTT assay and study on the reduction rate for LPS-induced inflammation, and details are provided within the manuscript. Specifically, you can find the explanation in lines 97~102:

“To investigate the potential protective effects of SPM against LPS-induced inflammation, we exposed cells to LPS at concentrations of 25 or 50ug/mL for 24h, followed by treatment with 10% or 20% SPM for an additional 24h. Different SPM doses (rang-ing from 10% to 50%) were previously evaluated in vitro, and 20% SPM showed the highest viability. Therefore, it was selected as the optimal dose for cell culture”

And in lines 106~110:

“To investigate the effect of SPM on inflammation, we assessed the IL-1b mRNA expression in HaCaT cells after a 24 h stimulation with LPS, followed by treatment with 10% or 20% SPM. The objective was to compare the optimal dose (20%) with the lowest dose (10%) of SPM to determine whether even the lowest dose exhibits an anti-inflammatory effect.”

Reviewer 2 Report

The manuscript is very well written. Thank you for making it so easy to read and understand!

Figures and images need major improvement. Font within figures is too small.

Figures are too small. Why not make them the same size as the body text?

TSLP abbreviation is not defined.

Fig 3. NOT acceptable. Very messy. Keep bars the same size but spread them apart, so that you have more space to write the description underneath. Consider C, for control. PIC/TNF, PIC/TNF+S and PIC/TNF+M and place these under each of the 4 bar graphs. You can explain the meanings of each abbreviation in the figure legend.

Why did you choose 3 weeks of treatment with SPM? What is the rationale or justification for 3 weeks? Cell turnover, repair, length of inflammatory response?

Fig 6 is not convincing? Only 1 patient?

Page 7, line 260: You mention enhancement of cellular metabolism. What about cellular repair.

Page 9, lines 333-334: Chronic eczema, xerosis cutis, atopic dermatitis, seborrheic dermatitis are all different entities, and this brings in too many variables. You should have 5 patients with each condition. You cannot include atopic dermatitis and seborrheic dermatitis in a single group. These conditions are different.

What are the races of the patients in this study? Demographic data could be improved by including chronic conditions (hypertension, diabetes, anemia, cancer, etc.), baseline levels of inflammatory markers, etc.

Author Response

Thank you very much for your precious comments that helped us improve this manuscript. Below we provide the point-by-point responses. 

1) Figures and images need major improvement. Font within figures is too small.

A: Thank you for highlighting the need to improve our figures and increase the font size. We recognize the importance of clear and accessible visual representation. In response to your comment, we have increased the font size within all figures, enhanced the resolution and clarity of the images, and made adjustments to the layout and design. These changes are aimed at improving readability and overall visual quality. We appreciate your feedback and believe the revised figures will address your concerns.

2) Figures are too small. Why not make them the same size as the body text?

A: Thank you for the suggestion to increase the size of the figures to match the body text. In our revised manuscript, we have resized the figures in pages 3~7 to align with the size of the body text, ensuring that the figures are clear and easily interpretable. We appreciate your insightful feedback and believe that this change enhances the manuscript's overall quality.

3) TSLP abbreviation is not defined.

A: We recognize the omission of the definition for the abbreviation "TSLP" in the original manuscript and apologize for this oversight. In the revised version, "TSLP" has been clearly defined as "Thymic Stromal Lymphopoietin" at its first mention in line 147:.

“We evaluated SPM’s anti-inflammatory effects by measuring the cytokine levels (IL-1β, IL-6, IL-33, and TSLP; Thymic stromal lymphopoietin) in HaCaT cells stimu-lated by poly(I:C)/TNFα (Figure 3a, 3b).”

4) Fig 3. NOT acceptable. Very messy. Keep bars the same size but spread them apart, so that you have more space to write the description underneath. Consider C, for control. PIC/TNF, PIC/TNF+S and PIC/TNF+M and place these under each of the 4 bar graphs. You can explain the meanings of each abbreviation in the figure legend.

A: We acknowledge the feedback regarding Figure 3's appearance and understand that its presentation was not optimal. Based on your valuable suggestions, we have made specific changes to improve clarity.

Revised Figure Legends in line 161-168:“Figure 3. SPM suppresses the expression of poly(I:C)/TNFα-induced cytokines in HaCaT cells. (a) Schematic experimental design of 24h poly(I:C)/TNFα pro-inflammatory cocktail stimulus ac-companied by SPM or Mometasone (positive control). (b) IL-1β, IL-6, IL-33, and TSLP mRNA levels after 24h poly(I:C)/TNFα stimulation and co-treatment with SPM or Mometasone. N=6 biological replicates. Error bars represent the mean ± SD. *p< 0.05; **p< 0.01; *** p< 0.001; **** p< 0.0001, as determined by One-way ANOVA followed by Tukey’s multiple comparison test. Ctr; control, P(I:C)/TNFα; poly(I:C)/TNFα + vehicle(water), P(I:C)/TNFa + S; poly(I:C)/TNFα + 20% SPM, P(I:C)/TNFα + M; poly(I:C)/TNFα + 1uM Mometasone.”

5) Why did you choose 3 weeks of treatment with SPM? What is the rationale or justification for 3 weeks? Cell turnover, repair, length of inflammatory response?

A: The 3-week treatment period with SPM was chosen based on biological considerations as the reviewer pointed out. The average turnover time for epidermal cells is around 28 days, so a 3-week duration allowed for observing changes in inflammation and regeneration. This time frame aligns with existing studies comparing moisturizers against topical steroids or tacrolimus in managing atopic dermatitis, often conducted over 3 to 4 weeks.

Additionally, the 3 or 4-week period aligns with common clinical practice for evaluating the effectiveness of topical treatments such as emollients, as demonstrated in comparative studies evaluating moisturizers against topical steroids or tacrolimus in the management of atopic dermatitis, which is a prototype disease model for chronic eczema. For example, a recent study by Shim et al, Dermatologic Therapy 2023 (https://www.hindawi.com/journals/dth/2023/4811165/) involving atopic dermatitis patients compared a moisturizer to topical tacrolimus over 4 weeks, revealing essential roles in restoring skin barrier function and maintenance therapy.

Importantly, the study was conducted without topical steroids, reflecting how SPM could be used in a real-world setting. The 3-week period was designed to capture the inflammatory response in chronic skin conditions like eczema and provide sufficient time for a comprehensive assessment of potential adverse reactions and changes in clinical symptoms, making it the optimized interval for this study.

6) Fig 6 is not convincing? Only 1 patient?

A: We acknowledge the concern that including only one patient's data might seem limited; however, we faced practical challenges in conducting the clinical study. To assess the histological changes within patients, it was necessary to take multiple skin biopsies both at baseline (from the control and test sites) and after 3 weeks of treatment (again from control and test sites). Conducting serial skin biopsies is an invasive procedure, and securing patient consent for this process proved challenging. Despite our efforts to involve more participants, we encountered a significant refusal rate, resulting in the inclusion of only one patient's biopsy in the figure.

This single biopsy, therefore, was included to provide an illustrative example of the changes observed during the treatment with SPM. Though it does not offer a comprehensive statistical view, it gives qualitative insight into the treatment's effects at the cellular level.

We, the authors, also feel regretful about this part. If the reviewer thinks this single example cannot be representative, we are willing to revise the figure as 'supplementary data.' This compromise would maintain the integrity of the manuscript while accommodating concerns about the representativeness of the data.

7) Page 7, line 260: You mention enhancement of cellular metabolism. What about cellular repair.

A: Thank you for bringing this to our attention. We agree with your observation regarding the importance of cellular repair. In our revised manuscript, we have included the term "cellular repair" to more accurately reflect this aspect of the study. We appreciate your valuable feedback and revised line 266-267:

“Currently, the best explanation is that vitamins play a non-specific permissive role by enhancing cellular repair.”

8) Page 9, lines 333-334: Chronic eczema, xerosis cutis, atopic dermatitis, seborrheic dermatitis are all different entities, and this brings in too many variables. You should have 5 patients with each condition. You cannot include atopic dermatitis and seborrheic dermatitis in a single group. These conditions are different.

A: We appreciate your concern regarding the inclusion of various conditions under the umbrella term 'eczematous skin lesions,' including chronic eczema, xerosis cutis, atopic dermatitis, and seborrheic dermatitis. We do recognize that these entities are different and may introduce variability. In our study design, we intentionally focused on the common inflammatory and symptomatic features shared across these conditions, as our primary goal was to evaluate the effectiveness of SPM on eczematous skin lesions rather than individual diagnoses. The rationale was to explore the broader application of SPM, reflecting a more generalized approach to treating the inflammation underlying these conditions. However, we understand the importance of specificity in clinical research and the need to consider the unique aspects of each condition.

We acknowledge that grouping various conditions into a single category might oversimplify their complexity. In the second group of our study, among six patients, four had a previous diagnosis of atopic dermatitis. We have recognized this distinction and have revised the table on page 9 accordingly.

We believe this generalized approach does not undermine the value of our findings, but we take your feedback seriously and will ensure to consider it in future studies. Thank you for your insightful comment.

9) What are the races of the patients in this study? Demographic data could be improved by including chronic conditions (hypertension, diabetes, anemia, cancer, etc.), baseline levels of inflammatory markers, etc.

A: We agree with the reviewer’s suggestion to enhance the demographic data in our study. In response to your point, we would like to clarify that all the patients in this study were of Korean ethnicity and were characterized as Fitzpatrick skin types II and III.

Additionally, we included the patients without any significant underlying medical history, such as hypertension, diabetes, anemia, cancer, etc., and we did not conduct tests for baseline inflammatory markers.

Additionally, we have revised the table on page 9, as per your previous question, to include this detailed information. We believe this revisions provide a more comprehensive understanding of our study population and sincerely thank you for bringing this to our attention.

Reviewer 3 Report

This work described the multi-component formulation for reducing the inflammation and chronic eczematous skin. The experiments were designed rationally and unfolded adequately. I recommend the minor revision on the following points.

1. The characteristics of the multi-component formulation should be detailed. If there was a commercial secret, at least the authors should provide a rough proportion of the main categories. This was the basis of the corresponding function.

2. I recommend some preliminary experiments on the inhibition of melanoma or the corresponding proteins. This kind of investigation can improve the depth of the skin-related research.

3. Please improve the language use.

Minor editing is required.

Author Response

Thank you very much for your precious comments that helped us improve this manuscript. Below we provide the point-by-point responses. 

1) The characteristics of the multi-component formulation should be detailed. If there was a commercial secret, at least the authors should provide a rough proportion of the main categories. This was the basis of the corresponding function.

A: Thank you for your valuable insight regarding the details of our multi-component formulation. In this study, we have chosen a method that balances the necessity for transparency with the protection of proprietary information. Specifically, we have arranged the order of the ingredients according to their volume, reflecting their relative abundance in the formulation. As detailed in lines 46~54 of our manuscript, the ingredients are listed in descending order, beginning with the most abundant.

We understand the importance of providing sufficient information about the composition, and we believe that this approach offers a general understanding without compromising any confidential aspects of the formulation. Your suggestion has certainly encouraged us to present this information in the most responsible and informative manner, and we sincerely appreciate your thoughtful feedback.

2) I recommend some preliminary experiments on the inhibition of melanoma or the corresponding proteins. This kind of investigation can improve the depth of the skin-related research.

A: That’s an excellent suggestion. Our research focus extends beyond anti-inflammation, involving a comprehensive exploration of various effects, including skin barrier strengthening and wound healing. We plan to publish this relevant information soon. Our research aims to provide a comprehensive understanding of the multifaceted benefits of our formulation and its potential applications in diverse skin-related conditions. Inhibition of melanoma is an intriguing idea, and we are enthusiastic about the potential to study this aspect further.

3) Please improve the language use.

A: Thank you for bringing attention to the language use in our manuscript. We have carefully reviewed and revised the text to ensure clarity, coherence, and adherence to proper English language conventions. Your feedback has helped enhance the quality of our work, and we sincerely appreciate it.